# Metabolic profiling during COVID-19 infection in humans: Identification of potential biomarkers for occurrence, severity and outcomes using machine learning

**Gamalat A. Elgedawy[1], Mohamed Samir[2]\*, Naglaa S. Elabd** [3]\***, Hala H. Elsaid[1], Mohamed Enar[4], Radwa H. Salem[5], Belal A. Montaser[6], Hind S. AboShabaan[7], Randa M. Seddik[3], Shimaa M. El-Askaeri** [5]**, Marwa M. Omar[6], Marwa L. Helal[1]**

1 Department of Clinical Biochemistry and Molecular Diagnostics, National Liver Institute, Menoufia University, Shebin El-Kom, Menoufia, Egypt, 2 Faculty of Veterinary Medicine, Department of Zoonoses, Zagazig University, Zagazig, Egypt, 3 Faculty of Medicine, Department of Tropical Medicine, Menoufia University, Shebin El-Kom, Menoufia, Egypt, 4 Al Mahala Elkobra Fever Hospital, Al Mahala Elkobra, Egypt, 5 Department of Clinical Microbiology and Immunology, National Liver Institute, Menoufia University, Shebin El-Kom, Menoufia, Egypt, 6 Faculty of Medicine, Department of Clinical Pathology, Menoufia University, Shebin El-Kom, Menoufia, Egypt, 7 Ph.D. of Biochemistry, National Liver Institute Hospital, Menoufia University, Shebin El-Kom, Menoufia, Egypt

\* naglaa.alabd.12@med.menofia.edu.eg, naglaa_elabd@yahoo.com (NE); mohsamir2016@yahoo.com (MS)

## Abstract

### Background

After its emergence in China, the coronavirus SARS-CoV-2 has swept the world, leading to global health crises with millions of deaths. COVID-19 clinical manifestations differ in severity, ranging from mild symptoms to severe disease. Although perturbation of metabolism has been reported as a part of the host response to COVID-19 infection, scarce data exist that describe stage-specific changes in host metabolites during the infection and how this could stratify patients based on severity.

### Methods

Given this knowledge gap, we performed targeted metabolomics profiling and then used machine learning models and biostatistics to characterize the alteration patterns of 50 metabolites and 17 blood parameters measured in a cohort of 295 human subjects. They were categorized into healthy controls, non-severe, severe and critical groups with their outcomes. Subject's demographic and clinical data were also used in the analyses to provide more robust predictive models.

### Results

The non-severe and severe COVID-19 patients experienced the strongest changes in metabolite repertoire, whereas less intense changes occur during the critical phase. Panels of 15, 14, 2 and 2 key metabolites were identified as predictors for non-severe, severe,

**Funding:** The UPLC MS/MS instrument was a grant from Science and Technology Development Fund (STDF), Egypt. Grant number: N2338. This funding enabled the establishing of the UPLC MS/MS instrument in our laboratory. We have used this instrument to measure the metabolite concentrations in this study.

**Competing interests:** The authors have declared that no competing interests exist.

critical and dead patients, respectively. Specifically, arginine and malonyl methylmalonyl succinylcarnitine were significant biomarkers for the onset of COVID-19 infection and taur-oursodeoxycholic acid were potential biomarkers for disease progression. Measuring blood parameters enhanced the predictive power of metabolic signatures during critical illness.

## Conclusions

Metabolomic signatures are distinctive for each stage of COVID-19 infection. This has great translation potential as it opens new therapeutic and diagnostic prospective based on key metabolites.

## Introduction

Since its first emergence in Wuhan city, China in December 2019, severe acute respiratory syndrome coronavirus 2 (SARS-CoV-2) continues to be a public health threat and is still spreading around the world, especially with new variants [1,2]. As of May 2023, more than seven hundred million cases were confirmed by the WHO with ~ 7 million were confirmed deaths globally [3]. The infection with such virus has greatly imposed financial and social burden for countries and individuals throughout the world attributed to its disastrous consequences on both patients and their families [4]. Egypt has been one of the countries that was impacted by COVID-19 pandemic, with up-to-date number of cases slightly exceeding half million, of those patients ~ 5% died from the disease [3].

Most individuals who acquire SARS-CoV-2 infection experience mild disease. However, estimates of ~ 14% of COVID-19 patients could develop severe disease and ~ 5% experience shock or multiple organ failure, as well as lung dysfunction necessitating ventilation, especially those with complications or those suffering acute respiratory distress syndrome (ARDS) [5]. With the virus trying to hijack the host machinery for survival purposes, the infection with SARS-CoV-2 triggers a plethora of host responses [6], of which metabolome represents one important component. The repertoire of human metabolomes represents an ensemble of several thousands of molecules that cover an amble range of concentration (from <1 nM to >1 μM) and are produced by either the host genome or the genome of host microflora [7]. The blood is the primary carrier of host metabolites, the relative concentration of which mirrors the patho-physiological status of an individual, and thus could inform virus-induced tissue lesions and organ failure. It has been reported that an organism's metabolome is a more accurate gauge of its metabolic state than its proteome or transcriptome [8].

Changes in endogenous metabolites is one of the characteristics of COVID-19 infection [9–11]. The intensive hypoxia associated with COVID-19-induced lung impairment possibly leads to altered metabolic profile [12]. Reports have indicated reductions in levels of some metabolites in severe COVID-19 infection in patients suffering from diabetes or hypertension [13]. Therefore, investigating the alteration in metabolites during COVID-19 infection could unravel important aspects of disease mechanisms such as revealing diagnostic or prognostic metabolic markers and discriminating patient groups based on disease severity [10,14,15]. Along with metabolite changes, hyperinflammation and over production of cytokines were observed during COVID-19 infection and has been recognized as a major cause of mortality in COVID-19 patients [16,17].

There have been multiple studies that profiled metabolites in COVID-19 patients aiming to seek novel biomarkers or stratifying the patients. However, most of these studies did not reflect

on the stage-to-stage differential regulation of metabolites during disease progression. Indeed, these studies have either compared controls *vs.* patients at each of the severity stages [18], simultaneously compared between all stages [14], or only contrasted controls with COVID-19 +ve patients without stage definition [19,20]. Indeed, the genesis of COVID-19 is a multistep process that progresses over time [15,18]. It is generally assumed that different stages of virus replication cycle from entry to virus release are entirely fueled by host's cell energy and metabolic resources [21]. This has already been demonstrated for SARS-CoV-2 infection in a robust animal model that mimics humans [22] and is published in the most recent WHO guidelines of patient stratification [23]. This already suggests that stage-to-stage reprogramming in host metabolites during COVID-19 infection is valid and worth investigated [18]. The accurate classification of patients group, however still challenging, particularly because of the wide and overlapped spectrum of patient's symptoms and, as a result, different pathophysiological pathways that are being affected and interrupted during disease progression. Although few metabolomics studies have used nuclear magnetic resonance (NMR) [24], the preferred method for exploring potentially diagnostic biomarkers in COVID-19 disease has been mass spectrometry (MS)-based metabolomics. Even though liquid chromatography (LC) coupled with MS has been used in many of these studies, gas chromatography and MS have been shown to produce intriguing results about the evolution of illness [25].

In the current study, a targeted metabolic analysis was applied on a cohort of Egyptian subjects who exemplified consecutive stages of COVID-19 infection with the purpose of determining the most promising metabolites that could be used as biomarkers for disease occurrence, progression, and outcome. We also tested, by data analytics and machine learning models, how this could inform patient disease stage and whether the addition of blood indices measured on the same subjects could substantiate the clinical biomarker utility of the identified metabolites.

## Material and method

### 2.1 Study design and study participants

From November 2021 to May 2022, 200 patients with confirmed COVID-19, who were admitted to Menoufia University Hospitals, Menoufia province, Egypt, were recruited. All patients had a clinical suspicion of COVID-19 and were confirmed to be positive using PCR applied on nasopharyngeal and oropharyngeal samples. The study subjects (initial number = 300) were categorized into 4 groups (100 healthy control (HC), 100 non-severe, 50 severe and 50 critical). This classification follows the last updates of WHO "COVID-19 Clinical management: Living guidance, 25 January 2021" [23]. The non-severe COVID-19 patients were described as having neither severe nor critical COVID-19 criteria. The severe COVID-19 included patients with any of the following criteria: Oxygen saturation < 90% on room air; in adults, signs of severe respiratory distress (accessory muscle use, inability to complete full sentences, respiratory rate > 30 breaths per minute), in addition to the signs of pneumonia. The critical COVID-19 patients should meet the criteria of acute respiratory distress syndrome (ARDS), sepsis, septic shock or other conditions that would normally require the provision of life-sustaining therapies such as mechanical ventilation (invasive or non-invasive) or vasopressor therapy. Patients with known history of hepatic, renal, cardiac diseases and coagulation disorders were excluded from the study. The HC included apparently healthy healthcare workers with no evidence of COVID-19 infection by standard clinical criteria and laboratory investigation. Patient's demographic and clinical data were also included in the study for analysis (**Table 1**).

**Ethical approval:** The study was conducted in accordance with Helsinki Declaration and was approved by National Liver Institute Ethical Committee (IRB: NLI 00003413). Prior to

**Table 1. Summary of demographic and clinical data of healthy individuals and COVID-19 patients.**

| | | Control (n = 99) | Non-severe (n = 100) | Severe (n = 46) | Critical (n = 50) | P-values** |
|---|---|---|---|---|---|---|
| **Age (median, Q1,Q3)** | | 39 (Q1: 38, Q3: 43) | 43.5 (Q1: 35, Q3:47.2) | 54.5 (Q1: 35.2, Q3: 60) | 70 (Q1: 43, Q3: 80) | <0.0001 |
| **Gender** | Female | 49(%49.4*) | 58(%58) | 36(%78.2) | 27(%54) | 0.0002 |
| | Male | 50(%50.5) | 42(%42) | 10(%21.7) | 23(%46) | |
| **CO-RADS score** | 0 | NA | 12(%12) | 0(%0) | 0(%0) | 1.24E-14 |
| | 3 | NA | 3(%3) | 0(%0) | 0(%0) | |
| | 4 | NA | 61(%61) | 6(%13) | 9(%18) | |
| | 5 | NA | 24(%24) | 40(%86.9) | 41(%82) | |
| **Severity score** | 0 | NA | 12(%12) | 0(%0) | 0(%0) | 2.20E-16 |
| | 1 | NA | 61(%61) | 5(%10.8) | 0(%0) | |
| | 2 | NA | 27(%27) | 20(%43.4) | 30(%60) | |
| | 3 | NA | 0(%0) | 21(%45.6) | 20(%40) | |
| **Symptoms** | Fever | NA | 73(%73) | 41(%89.1) | 43(%86) | 0.02446 |
| | Dry cough | NA | 90(%90) | 30(%65.2) | 28(%56) | 8.03E-06 |
| | Productive Cough | NA | 10(%10) | 16(%34.7) | 22(%44) | 8.03E-06 |
| | Dyspnea | NA | 71(%71) | 37(%80.4) | 42(%84) | 0.1273 |
| | Myalgia | NA | 57(%57) | 17(%36.9) | 24(%48) | 0.02897 |
| | Bone ache | NA | 61(%61) | 20(%43.4) | 24(%48) | 0.04003 |
| | Anosmia | NA | 54(%54) | 32(%69.5) | 36(%72) | 0.03096 |
| | Loss of taste | NA | 19(%19) | 18(%39.1) | 19(%38) | 0.002538 |
| | Vomiting | NA | 12(%12) | 6(%13) | 7(%14) | 0.9337 |
| | Diarrhea | NA | 54(%54) | 33(%71.7) | 36(%72) | 0.01993 |
| | Conjunctivitis | NA | 3(%3) | 7(%15.2) | 23(%46) | 1.66E-10 |
| | Runny nose | NA | 70(%70) | 26(%56.5) | 36(%72) | 0.3638 |
| **Comorbidities** | Diabetes (DM) | NA | 46(%46) | 26(%56.5) | 38(%76) | <0.0001 |
| | Hypertension (HT) | NA | 13(%13) | 13(%28.2) | 31(%62) | <0.0001 |
| | Diabetes plus hypertension | NA | 10(%10) | 13(%28.2) | 27(%54) | <0.0001 |
| | No comorbidity | NA | 51(%51) | 20(%43.4) | 8(%16) | 0.0002 |
| **Outcome** | Died | NA | 0(%0) | 10(%21.7) | 37(%74) | <0.0001 |
| | Survived | NA | 100(%100) | 36(%78.2) | 13(%26) | |

*The percentages mentioned in the table represent the proportion of subjects having particular gender, score, symptoms out of the total number of subjects in the respective COVID-19 group.

**P-values of numeric parameters were calculated by using two-tailed student t test with assumption of unequal variance. P-values of categorical parameters were calculated using a chi-square test (or Fisher's exact test when suitable).

enrollment, each participant was informed about the aims of the study and was offered the chance to sign their informed written consent.

## 2.2 Clinical assessment and samples collection

Patients with clinical suspicion of COVID-19 were assessed upon attending to the COVID-19 isolation unit at the Faculty of Medicine, Menoufia University Hospital throughout the research period. Following assessments by clinical, laboratory, and radiological means, patients were classified as non-severe, severe, or critical. For cases that were considered to be non-severe, an outpatient treatment prescription was provided, and further follow-up was conducted over the phone or in the COVID-19 outpatient clinic. Those with severe or critical presentations were admitted to the COVID-19 quarantine ward or ICU, where baseline clinical,

laboratory, and radiological data were recorded at the time of admission. Additionally, a daily evaluation of the course of the illness, the response to treatment were evaluated and recorded. Blood samples (10 ml) from all subjects were drawn from the cubital vein by venipuncture after fasting at the time of admission/diagnosis. Of these, 5 ml were collected in a plain vacutainer tube and allowed to clot at room temperature then centrifuged (3000 rpm, 5 min.) and the clear supernatant sera were separated and collected in 3- aliquots. The 1[st] serum aliquot was used to measure ferritin, procalcitonin, c-reactive protein (CRP), LDH (lactate dehydrogenase), liver enzymes (ALT, AST), and kidney function tests (urea and creatinine). The 2[nd] serum aliquot was used to measure IL-6 using human IL6 ELISA kit following manufacturer instructions purchased from Thermofisher Scientific, US (Catalog no. EH2IL6). The 3[rd] serum aliquot was kept at -80 for bile acids analysis. From the remaining 5 ml, 2ml were collected into a tube containing Ethylene diaminetetraacetic acid (EDTA) for the complete blood count (CBC), 1 ml was spotted on filter paper (903 Whattman paper, NJ, USA), left to dry on a clean surface for 6 hours, and then stored at $-80$ C˚ until analyzing amino acids, carnitine, and acyl carnitine and the final 2 ml were collected into sodium citrate tubes for D-dimer, prothrombin time, and INR measurements.

## 2.3 Targeted metabolomics using ultra-performance liquid chromatography tandem mass spectrometry (UPLC MS/MS)

**2.3.1 Amino acid and carnitines quantification.** Amino acid and blood carnitine and L carnitines were measured using MassChrom® Amino Acids and Acylcarnitines from Dried Blood / Non Derivatised—LC-MS/MS (order No.: 57000/F, Chromsystems Instruments & Chemicals GmbH, Germany). Three mm of the dried blood spot disk was punched into a well of the v-bottomed plate containing 100 µl of lyophilized internal standard reconstituted with 25 ml Extraction Buffer. The plate was sealed with a protective sheet and agitated at 600 rpm for 20 min at room temperature. The supernatant was transferred to a new v-bottomed well plate and covered by aluminum foil sheet. Ten µl of the elute was injected into the MS/MS system at a two-min interval in a flowing stream of 80% acetonitrile at a flow rate of 200 µl/min and reduced to 20 µl/min in 0.25 min. The flow rate increased to (600 µl/min in 1.25 min) then decreased again to (200 µl/min). The scan time of the MS/MS system was 1.2 min. The obtained spectra of all analytes analyzed with multiple reactions monitoring (MRM) mode. The quantitative analysis was achieved using Neolynx software (Neolynx Inc., Glendale, CA, USA) by comparing the signal intensity of an analyte against the corresponding internal standard. The quantification included 14 blood amino acids, 26 blood carnitine and L carnitines (S1 Table).

**2.3.2 Bile acids quantification.** We used standards for the bile acids listed in S1 Table. These were purchased from Sigma-Aldrich Chemicals (Merck KGaA, Darmstadt, Germany). Sample preparation for bile acid quantification was done according to [26] with modification. First, 100 µL blood sample was added to 400 µL ice-cold methanol to precipitate the sample proteins. The mixture was then vortexed, centrifuged (13500 rpm for 15 minutes) and the supernatant was obtained and centrifuged (13500 rpm for 15 minute). Finally, 50 µL of the final supernatant was mixed with 100 µL water/formic acid (1000: 1, v/v). The solution was then injected into LC/MS/MS system. Stock solution of 14 individual bile acids standards were dissolved separately in methanol to form of 10 mmol/L, and then stored at $-20$˚C [26]. The individual stock solutions were then pooled together to obtain mixture of 50 µmol /L in (50:50) deionized water and acetonitrile. Eight-point standard solutions ranging from 0, to 40 µmol/L were prepared by adding appropriate amounts of the mixture 50 µmol /L solution into the bile acid free pooled serum for external standard calibration. 5µmol/L of individual

stock solutions was injected to LC. Chromatographic separation was carried out on a triple-quadruple tandem MS. The analytical column was ACQUITY UPLC BEH C18, 1.7 μm, 2.1x50mm, column (Waters) at 50˚C. 5 μL of samples were eluted with a gradient at a flow rate of 0.28 mL/min. Mobile phase A was water/formic acid (1000: 1, v/v) and mobile phase B was acetonitrile. The elution started with 80% mobile phase A and 20% mobile phase B for an initial 2.1 min after injection, then with a linear gradient of mobile phase B of 20% to 30% over 5.2 min, followed by mobile phase B at 80% over 8 min, which was held for 0.5 min. the column was equilibrated with 80% mobile phase A for 2 min before the injection of the next sample.

## 2.4 Data analysis and machine learning models

The overall study design is shown in **Fig 1**. Initially, 54 metabolites (14 serum bile acids, 14 blood amino acid, 26 blood carnitine and L carnitines) that were measured in 300 subjects (100 HC, 100 non-severe, 50 sever and 50 critical) were included in the analyses. Principle component analyses (PCA) was used to visualize outliers in subjects and metabolites. Outliers were subsequently detected using boxplot with whisker. These analyses were done using base R functions in R software version 4.3.1. The raw concentration of metabolites was median normalized, log-transformed and then scaled by mean centralization divided by standard deviation of each variable. The normalized counts were used for further analyses. To compare metabolites concentration among and within study groups, we run 3- machine learning models; Partial least square discriminate analyses (PLS-DA), its orthogonal version (oPLS-DA) and random forest (RF) models. The parameters used to evaluate the performance of the PLS-DA model were accuracy, Q2 (classification ability) and R2 (predictability) [27]. These were generated with 3-component (to avoid over-fitting) and 10-fold cross validation. The performance of the oPLS-DA model was assessed using R2Y (variance among subjects as explained by the model) and Q2Y (model predictability) parameters. The significance of the oPLS-DA model was assessed at 0.5 for R2Y and Q2Y [28,29]. The RF model was applied using an ensemble of 1000 trees. The mtry function (R software version 4.3.1, random forest package [30]) was used

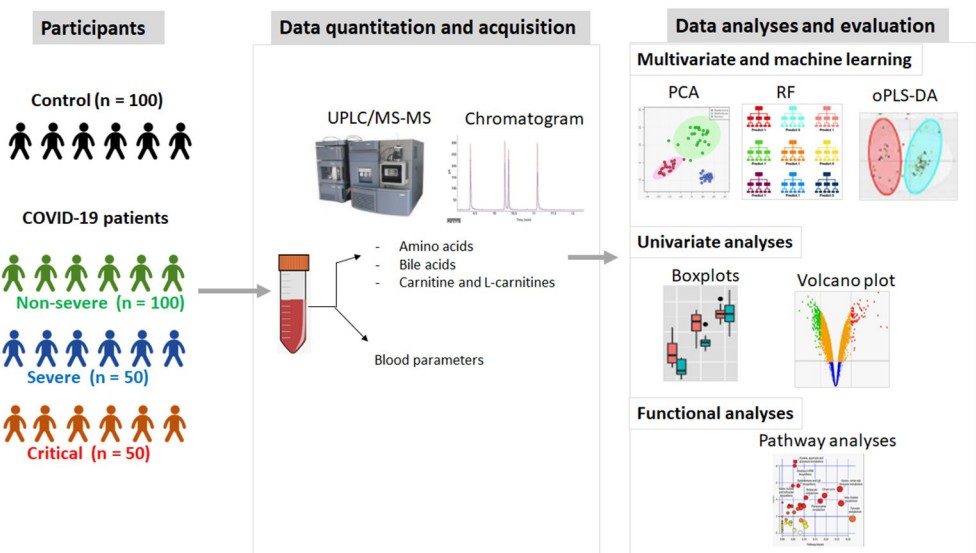

**Fig 1. Schematic diagram showing the overall design of the study.** The general steps were subject selection and stratification, data quantification, acquisition, data analyses, and model evaluation.

to set the reliable number of variables randomly sampled as candidates at each split (square root of number of variable). For univariate analyses, data normality was calculated using Shapiro wike test [31], and student *t*-test was run to obtain significant differences between pairs of groups. One-way ANOVA (or its non-parametric mate Kruskal-Wallis test) was used whenever needed after correction for multiple comparison testing using Dunn's test. Adj. P-value <0.05 was considered significant. To obtain the degree and pattern of differential expression in metabolites, data (non-normalized for features) were used to calculate the ratio between each group pairs. A fold change value of |1.5| and adj. P-value of 0.05 was used as cutoffs for significance. These data were visualized using volcano plots generated in R software version 4.3.1, package ggplot2 [32]. The finally selected panel of key diagnostic or prognostic metabolites should meet the following criteria: Fold change value of 1.5 (using only those that are significant with adj. P-value < 0.05), oPLS-DA- revealed VIP score > 1 and should be among top 15 metabolites as revealed by the RF model (based on the mean decrease in accuracy). ROC curves for comorbidities were visualized as fraction and its confidence intervals (CI) were estimated using Wilson/Brown method, and significant AUC was set at a cutoff = 0.05. Pathway enrichment analyses were done using Metaboanalyst v.5 [33] and is based on combined functional enrichment analyses and network topology. Hypergeometric test was used for functional enrichment analyses. Significant pathways were selected based on adj. P-value = 0.05.

## 3. Results

### 3.1 Exploratory data analyses

To exclude unwanted variation that could bias the analyses, PCA plot based on whole metabolites, metabolites subclasses, blood indices and the subjects were used to identify potential outliers. The analyses revealed some outliers within HC and severe groups. To more accurately identify outlier subject, mean normalized counts of metabolites in each class were visualized using boxplot with whiskers. Based on these analyses, 5 subjects (4 severe covid-19 patients based on amino acid profile and 1 HC based on the bile acid profile) were excluded. Other carnitines were also excluded because they were not detected in all subjects (e.g. tetradecanoyl carnitine, tetradecadienoyl and 3-hydroxyoctadecenoylcarnitine) or because it gave no signals in 94.3% (n = 283) of the subjects and had very low concentration in the remaining subjects (e.g. 3-Hydroxyhexadecanoyl). After outliers' removal, a total of 295 subjects (99 HC, 100 nonsever, 46 sever and 50 critical), 50 metabolites (14 amino acids, 14 bile acids and 22 carnitines) and 17 blood parameters entered the formal analysis (**S1 Table**). The counts of metabolite and their subclasses were successfully normalized (**S1 Fig**).

### 3.2 Demographic, clinical characteristics, laboratory values and disease outcomes of study participants

After outlier removal, the clinical cohort presented in this study consisted of 295 subjects, whose demographic and clinical characteristics are shown in **Table 1** and **S2 and S3 Figs**. The severe and critical patients have older participants than other groups, with their ages being significantly higher than those in the HC group (P-value < 0.0001). The whole cohort contained significantly (P-value = 0.0002) more females (n = 170, 57.6%) than males (n = 125, 42.4%) and female patients (n = 121) were significantly (P-value = 0.04) more than male patients (n = 75). Upon admission, the COVID-19 patients were evaluated for severity by COVID-19 Reporting and Data System (CO-RADS), and severity assessment. It was found that 53.5% of the patients had a CO-RADS score of 5, the majority of whom (77.1%) presented with critical (n = 41) or severe (n = 40) disease. All the COVID-19 patients were symptomatic, showing at

least one symptom (**S2F Fig**), with fever being the most represented symptom (80.1%) followed by dyspnea (76.5%) and dry cough (75.5%). The highest recorded symptoms in the non-severe group were dry cough and fever recorded in 90 and 73% of these patients, respectively, whereas both fever and dyspnea represented the top symptoms in both severe (89.1, 80.4%, respectively) and critical COVID-19 patients (86, 84%, respectively). The number of individuals with diabetes, hypertension, or both together varied significantly among all groups (P-value < 0.01) (**Table 1**). Out of the 196 COVID-19 patients, 40.3% (n = 79) had no comorbidities, 56.1% (n = 110) had diabetes, 29.1% (n = 57) had hypertension, and 25.5% (n = 50) had both diabetes and hypertension. The presence of these comorbidities varied according to COVID-19 severity. Half of non-severe (51%), close to half of severe (43.4%) and only 16% of critical COVID-19 patients had neither diabetes nor hypertension. The representation of diabetes was higher than hypertension in all severity groups with hypertension being non-reported in sever patients (**S3A Fig**). We observed an upward increase in the proportion of patients with concurrent diabetes and hypertension as the disease severity increases with those having concurrent diabetes and hypertension representing considerably high proportion of severe (28.2%) and critical (54%) COVID-19 patients. We were also able to follow the patient's outcomes. There was significant association between disease stage and the outcome (P-value = <0.0001). The majority (78.5%) of severe COVID-19 patients survived the infection, whereas the majority (74%) of the critical cases died (**S3B Fig**). Seventeen blood parameters were measured at the time of patient admission (**S2 Table** and **S4 Fig**). All blood parameters showed significant differences among the studied groups, in particular comparing HC and non-severe groups. However, the trends in among-group differences were characteristic for some parameters. For instance, inflammation-related indices (e.g. serum ferritin, CRP, LDH, procalcitonin, and D-dimer) showed an increase as the patients exhibited more severe disease, yet they exhibited a significant increase in severe compared to non-severe cases (**S2B Table**). IL-6 showed a significant increase in severe and critical cases over the HC but was none significantly altered between severe and critical patients. Furthermore, lymphocytes showed significant (P-value = 0.006) reduction in severe compared to non-severe patients, whereas creatinine showed the reverse pattern. INR was the only parameter that showed a significant increase in critical patients over the severe ones.

## 3.3 Diagnostic and prognostic machine learning models

Based on the normalized counts of all metabolites, the initial PCA revealed clear separation between HC and COVID-19 patients with various degrees of severity (i.e. non-severe, severe and critical patients), which rather appeared to overlap. This holds true when blood parameters are added to metabolites (**S5 Fig**). Comparable separation patterns were seen when the same analyses were applied to each metabolite subclass and blood parameters (**S6 Fig**). Comparing all groups, the PLS-DA model applied to the validation data reflected the results from the PCA, with moderate performance (accuracy: 0.67, R2: 0.7, Q2: 0.7, permutation test: P-value < 0.05) (**S7 Fig** and **Table 2**). When applied to the validation data, the RF classification model outperformed the PLS-DA model, producing a significantly higher classification accuracy of 0.97 (CI: 0.9–0.9), P-value < 0.0001, and misclassification error of only 0.02.

**3.3.1 Models to predict COVID-19 occurrence (distinguishing HC from non-severe patients).** To determine how changes in metabolites could discriminate HC from non-severe patients, who are at the early infection phase (i.e., suffering mild and moderate disease), we trained and validated several machine learning models on the data comparing these two categories. As shown in **S8A Fig** and **Table 2**, the PLS-DA gave a clear separation between both groups (accuracy = 1, R2 = 0.95, Q2 = 0.95 at the 1[st] component). Applying the oPLS-DA

**Table 2. Parameters for evaluation of each machine learning model discriminating pairs of groups using both the metabolites alone or metabolites plus blood indices.** NA: Not available. PLS-DAL partial least square discriminate analyses. oPLS-DA: Orthogonal partial least square discriminate analyses. RF: Random forest.

| Model performance | Among all groups | | Control vs non-severe | | Non-severe vs severe | | Severe vs critical | |
|---|---|---|---|---|---|---|---|---|
| | Metabolites | Metabolites + blood | Metabolites | Metabolites + blood | Metabolites | Metabolites + blood | Metabolites | Metabolites + blood |
| **Accuracy (PLS-DA)** | 0.67 | 0.67 | 1 | 1 | 0.96 | 0.95 | 0.63 | 0.82 |
| **R2 (PLS-DA)** | 0.7 | 0.7 | 0.95 | 0.92 | 0.76 | 0.63 | 0.47 | 0.54 |
| **Q2 (PLS-DA)** | 0.7 | 0.7 | 0.95 | 0.92 | 0.74 | 0.59 | 0.10 | 0.31 |
| **R2Y (oPLS-DA)** | NA | NA | 0.95 | 0.92 | 0.76 | 0.63 | 0.47 | 0.54 |
| **Q2Y (oPLS-DA)** | NA | NA | 0.95 | 0.92 | 0.73 | 0.61 | 0.15 | 0.24 |
| **Accuracy (RF)** | 0.97 | 0.97 | 1 | 1 | 1 | 1 | 0.92 | 0.92 |

model further supports this (R2Y and Q2Y = 0.95, P-value = 5.152e-16). As shown in **Fig 2A** and **Table 2**, adopting the RF model validated the results of the previous models, yet with better and significant differential clustering with accuracy, sensitivity, and specificity equal to 1. To exclude that this is an overfitting problem, we rerun the RF model using different proportions of training and validation data sets, which gave the same results. The predictability of this RF model was high as determined by ROC analyses (**Fig 2B**). To determine key metabolites that drive the separation between HC and non-severe groups, we combined the prediction results of RF, oPLS-DA models and univariate analyses. The oPLS-DA model predicted 18 metabolites with VIP score > 1 (**S3 Table**), the top of which were arginine, malonyl methylmalonyl succinylcarnitine, and tyrosine. The RF model determined key metabolites with high discriminatory ability as determined by their mean decrease in accuracy and GINI indices (values of both indices are listed for all metabolites in **S4 Table**). The top 15 metabolites that have the highest mean decrease in accuracy by RF mode are visualized in **Fig 2C**. The highest 3 metabolites by mean decrease in accuracy were malonyl methylmalonyl muccinylcarnitine (carnitines), glycodeoxycholic acid (bile acid) and arginine (amino acid). The first two of these were also the top metabolites based on mean decrease in GINI index (**S4 Table**). The results of univariate analyses are shown as volcano plot in **Fig 2D** and **S5 Table**. Out of the 50 metabolites, 37 (74%) were significantly differentially expressed (DE) between HC and non-sever groups (16 up regulated and 21 down regulated). These were almost evenly represented across metabolite categories (11 bile acids, 12 amino acids and 14 carnitines). Intersecting the results of RF, oPLS-DA models with univariate analyses generated a list of 15 metabolites that are important predictors for non-severe disease (Panel 1) (**Table 3**, **S9 Fig**). The metabolic pathways associated with these predictor metabolites are shown in **Table 4**. Aminoacyl-tRNA biosynthesis and valine, leucine and isoleucine biosynthesis were significantly enriched in this group. Phenylalanine, tyrosine and tryptophan biosynthesis pathway has the highest impact but was non-significantly enriched (adj. P-value >0.05).

**3.3.2. Models to predict COVID-19 severity.** *3.3.2.1 Models comparing non-severe and severe COVID-19 patients.* To reveal metabolites associated with severity and progression of COVID-19 infection, we first compared non-severe and severe groups. As shown in **S8b Fig** and **Table 2**, PLS-DA model generated significant separation between both groups (accuracy: 0.96, R2 = 0.76, Q2 = 0.74 at 1$^{st}$ component). In addition, oPLS-DA obtained comparable results (R2Y = 0.76, Q2Y = 0.73, P-value < 0.0001). Random forest model revealed perfect separation with an accuracy = 1 (P-value = 4.405e-08), 100% sensitivity and 100% specificity (**Fig 3A**). Applying RF on different proportions of training and validation data sets obtained similar results. This model was found to have excellent classification ability using ROC analyses (AUC = 1) (**Fig 3B**).

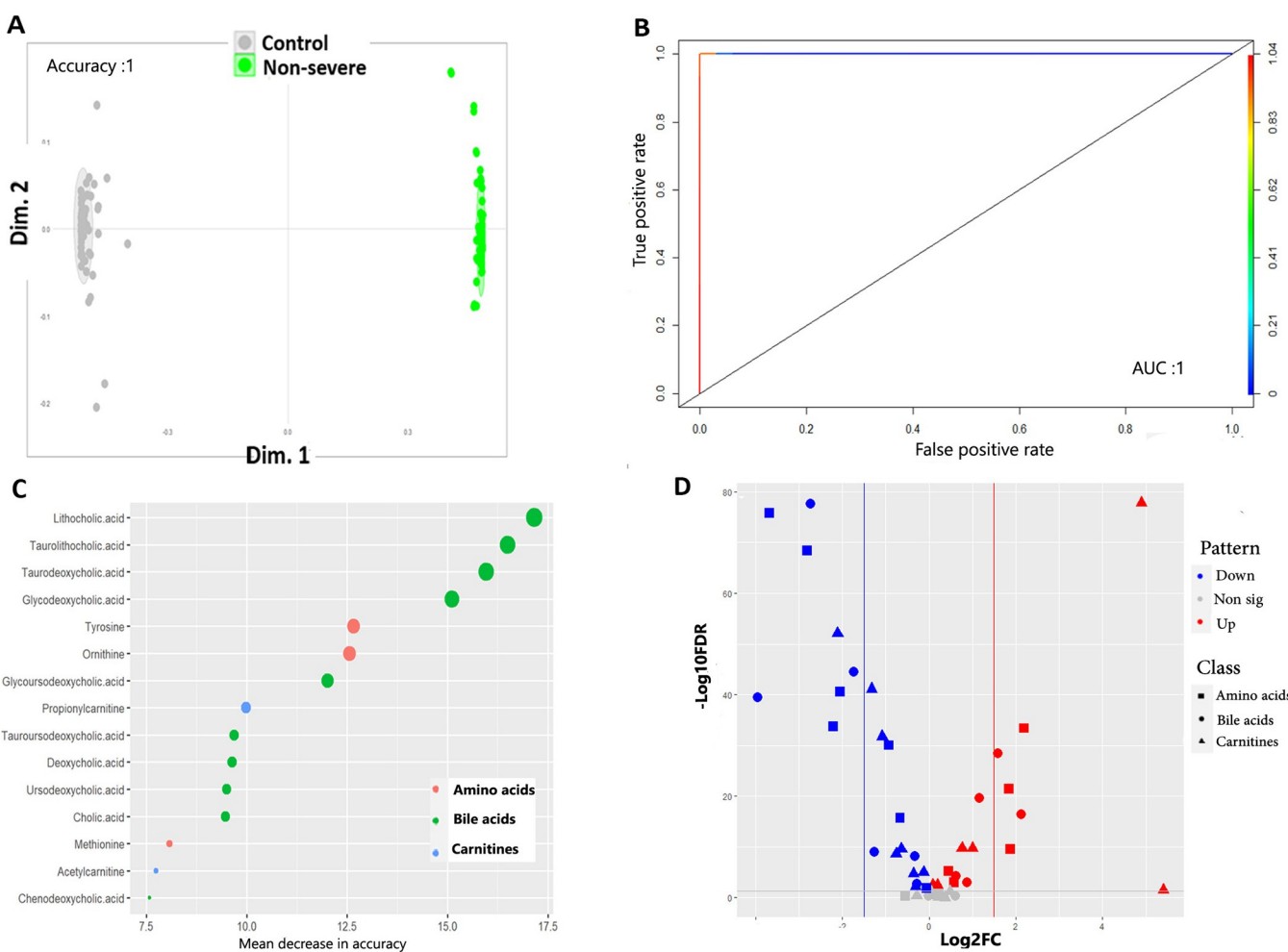

**Fig 2. Machine learning models and univariate analyses discriminating HC from non-severe COVID-19 patients. A.** Proximity plot of the RF model discriminating controls from non-sever COVID-19 patients. The ellipse shows confidence intervals **B.** ROC analyses showing the prediction ability of the model. AUC: Area under the curve. **C.** Top 15 metabolites that are important predictors for non-severe COVID-19 patients as revealed by RF model. The metabolites are color-grouped by their class and are ranked descending by their mean decrease in accuracy (the higher the mean decrease in accuracy the more important the metabolite). **D.** Volcano plot showing the results of the univariate analyses. The figure depicts the relationship between log2FC value of each metabolite (x-axis) against its -log10FDR (y-axis). Red and blue dots refer to metabolites that are significantly up and down regulated, respectively. Non-significant DE metabolites are shown as grey dots (-log10 adj. p value <1.3). The metabolite class are shape coded. The dashed horizontal line refers to the value of 1.3, the -log10 for a 0.05 FDR. The vertical dashed lines refer to the cutoff that equates a log2fold change value of |1.5|.

To obtain key metabolites with high prognostic value for severe disease, oPLS-DA model firstly revealed 20 prognostic metabolites with VIP score > 1, where glycodeoxycholic acid, taurodeoxycholic acid and tyrosine scored the top (Full list are in **S3 Table**). Applying RF models produced a list of key metabolites with high predictive value for severe disease (Full list are in **S4 Table**). The top 15 metabolites identified by RF model ranked by mean decrease in accuracy are shown in **Fig 3C**. The highest three of which were bile acids (nam+ely: lithocholic acid, taurolithocholic acid and taurodeoxycholic acid). Lithocholic acid and taurodeoxycholic acid were also the top ones as determined by mean decrease in GINI index. The univariate analyses performed on the same pairwise comparison identified 39 metabolites (78%) as being significant DE between non-severe and severe patients (Full list are in **S5 Table** and are shown **Fig 3D**). These included 27 up regulated and 12 down regulated metabolites. These significant DE metabolites were represented by 11 bile acids, 13 amino acids and 15 carnitines.

**Table 3. Final panel of key metabolites as determined by univariate analysis, random forest and oPLA-DA models.**

| | Metabolite class | Univariate analyses | | | Random forest model | oPLS-DA model |
|---|---|---|---|---|---|---|
| | | Log$^2$FC | - Log$^{10}$ FDR | DE Pattern | Mean decrease in model accuracy | VIP score |
| **HC vs non-severe (Panel 1)** | | | | | | |
| 3-Hydroxyoctadecanoylcarnitine | Carnitines | -1.1 | 31.7 | ↓ | 5.99 | 1.49 |
| Arginine | Amino acid | -3.7 | 75.7 | ↓ | 12.99 | 1.84 |
| Chenodeoxycholic acid | Bile acid | 1.6 | 28.4 | ↑ | 4.78 | 1.36 |
| Glutarylcarnitine | Carnitines | -1.3 | 41.1 | ↓ | 9.47 | 1.63 |
| Glycodeoxycholic acid | Bile acid | -2.7 | 77.6 | ↓ | 13.77 | 1.55 |
| Leucine | Amino acid | -2.0 | 40.5 | ↓ | 7.76 | 1.55 |
| Lithocholic acid | Bile acid | -1.7 | 44.5 | ↓ | 9.66 | 1.52 |
| Malonyl Methylmalonyl Succinylcarnitine | Carnitines | 4.9 | 77.8 | ↑ | 14.14 | 1.82 |
| Methionine | Amino acid | -2.2 | 33.7 | ↓ | 6.10 | 1.61 |
| Phe/Tyr ratio | Amino acid | 2.2 | 33.4 | ↑ | 6.08 | 1.30 |
| Proline | Amino acid | 1.8 | 21.5 | ↑ | 4.71 | 1.45 |
| Taurocholic acid | Bile acid | -4.0 | 39.4 | ↓ | 7.54 | 1.17 |
| Tiglyl 3-Methylcrotonyl | Carnitines | -2.1 | 52.0 | ↓ | 12.67 | 1.77 |
| Tyrosine | Amino acid | -2.8 | 68.4 | ↓ | 12.95 | 1.81 |
| Valine | Amino acid | -0.9 | 30.0 | ↓ | 5.20 | 1.48 |
| **Non-severe vs severe (Panel 2)** | | | | | | |
| Acetylcarnitine | Carnitines | 1.4 | 9.7 | ↑ | 7.74 | 1.44 |
| Chenodeoxycholic acid | Bile acid | 0.7 | 6.1 | ↑ | 7.58 | 1.15 |
| Deoxycholic acid | Bile acid | -1.1 | 4.9 | ↓ | 9.63 | 1.31 |
| Glycodeoxycholic acid | Bile acid | -2.7 | 15.7 | ↓ | 15.12 | 1.81 |
| Glycoursodeoxycholic acid | Bile acid | -1.7 | 10.3 | ↓ | 12.02 | 1.59 |
| Lithocholic acid | Bile acid | -1.3 | 8.3 | ↓ | 17.16 | 1.68 |
| Methionine | Amino acid | 0.9 | 6.6 | ↑ | 8.07 | 1.15 |
| Ornithine | Amino acid | 0.9 | 15.9 | ↑ | 12.56 | 1.53 |
| Propionylcarnitine | Carnitines | 1.4 | 10.2 | ↑ | 9.99 | 1.26 |
| Taurodeoxycholic acid | Bile acid | -3.6 | 12.3 | ↓ | 15.97 | 1.77 |
| Taurolithocholic acid | Bile acid | -1.7 | 7.9 | ↓ | 16.50 | 1.67 |
| Tyrosine | Amino acid | 1.4 | 19.0 | ↑ | 12.65 | 1.71 |
| Ursodeoxycholic acid | Bile acid | -0.8 | 5.4 | ↓ | 9.50 | 1.38 |
| Tauroursodeoxycholic acid | Bile acid | -0.17 | 3.42 | ↓ | 9.6 | 1.1 |
| **Severe vs critical (Panel 3)** | | | | | | |
| Tauroursodeoxycholic acid | Bile acid | -0.39 | 2.5 | ↓ | 12.7 | 2.5 |
| Malonyl Methylmalonyl Succinylcarnitine | Carnitines | 1.1 | 2.1 | ↑ | 14.44 | 2.39 |
| **Survival vs died (Panel 4)** | | | | | | |
| Hexanoylcarnitine | Carnitines | -0.847 | 2.3 | ↓ | 12.40 | 2.00 |
| Octenoylcarnitine | Carnitines | -0.825 | 1.3 | ↓ | 12.37 | 1.72 |

Combining the lists of metabolites that are predicted by RF, oPLS-DA models and univariate analyses obtained a panel of 14 metabolites that are important predictors for severe disease (Panel 2) (**Table 3** and **S9 Fig**). As shown in **Table 4**, the pathways enrichment analyses on those 14 metabolites revealed the presence of primary bile acid biosynthesis, tyrosine metabolism and arginine biosynthesis pathways as significantly enriched.

3.3.2.2 *Models comparing severe and critical COVID-19 patients*. To demonstrate the potential of metabolites as prognostic markers for critical COVID-19 patients, severe and critical groups were compared in the models. As shown in **S10A Fig and Table 2**, PLS-DA model

**Table 4. Pathways enriched by the predictor metabolites in each pairwise comparison.**

| Pathway Name | p | -log(p) | FDR | Impact |
|---|---|---|---|---|
| **HC vs non-Severe** | | | | |
| Phenylalanine, tyrosine and tryptophan biosynthesis | 0.0 | 1.7 | 0.3 | 0.5 |
| Tyrosine metabolism | 0.2 | 0.7 | 1.0 | 0.1 |
| Arginine and proline metabolism | 0.0 | 1.8 | 0.3 | 0.1 |
| Cysteine and methionine metabolism | 0.2 | 0.8 | 1.0 | 0.1 |
| Arginine biosynthesis | 0.1 | 1.2 | 0.6 | 0.1 |
| Primary bile acid biosynthesis | 0.0 | 1.7 | 0.3 | 0.0 |
| Aminoacyl-tRNA biosynthesis | 0.0 | 7.8 | 0.0 | 0.0 |
| Valine, leucine and isoleucine biosynthesis | 0.0 | 3.2 | 0.0 | 0.0 |
| Valine, leucine and isoleucine degradation | 0.0 | 1.8 | 0.3 | 0.0 |
| Taurine and hypotaurine metabolism | 0.0 | 1.4 | 0.5 | 0.0 |
| Ubiquinone and other terpenoid-quinone biosynthesis | 0.0 | 1.3 | 0.5 | 0.0 |
| Phenylalanine metabolism | 0.1 | 1.3 | 0.5 | 0.0 |
| Pantothenate and CoA biosynthesis | 0.1 | 1.0 | 0.7 | 0.0 |
| **Non-severe vs severe** | | | | |
| Phenylalanine, tyrosine and tryptophan biosynthesis | 0.0 | 2.0 | 0.4 | 0.5 |
| Tyrosine metabolism | 0.1 | 1.0 | 1.0 | 0.1 |
| Arginine and proline metabolism | 0.1 | 1.0 | 1.0 | 0.1 |
| Cysteine and methionine metabolism | 0.1 | 1.1 | 1.0 | 0.1 |
| Arginine biosynthesis | 0.0 | 1.4 | 0.6 | 0.1 |
| Aminoacyl-tRNA biosynthesis | 0.0 | 2.3 | 0.4 | 0.0 |
| Ubiquinone and other terpenoid-quinone biosynthesis | 0.0 | 1.6 | 0.5 | 0.0 |
| Phenylalanine metabolism | 0.0 | 1.6 | 0.5 | 0.0 |
| Glutathione metabolism | 0.1 | 1.2 | 1.0 | 0.0 |
| Primary bile acid biosynthesis | 0.1 | 0.9 | 1.0 | 0.0 |
| **Severe vs critical** | | | | |
| D-Glutamine and D-glutamate metabolism | 0.0 | 1.6 | 0.3 | 0.5 |
| Arginine biosynthesis | 0.0 | 2.8 | 0.0 | 0.3 |
| Glycine, serine and threonine metabolism | 0.1 | 0.9 | 0.9 | 0.2 |
| Alanine, aspartate and glutamate metabolism | 0.0 | 2.2 | 0.1 | 0.2 |
| Glutathione metabolism | 0.0 | 2.2 | 0.1 | 0.1 |
| Glyoxylate and dicarboxylate metabolism | 0.0 | 2.1 | 0.1 | 0.1 |
| Arginine and proline metabolism | 0.2 | 0.8 | 1.0 | 0.1 |
| Primary bile acid biosynthesis | 0.0 | 4.7 | 0.0 | 0.0 |
| Aminoacyl-tRNA biosynthesis | 0.0 | 3.0 | 0.0 | 0.0 |
| Porphyrin and chlorophyll metabolism | 0.0 | 2.1 | 0.1 | 0.0 |
| Nitrogen metabolism | 0.0 | 1.6 | 0.3 | 0.0 |
| Butanoate metabolism | 0.1 | 1.2 | 0.5 | 0.0 |
| Histidine metabolism | 0.1 | 1.2 | 0.5 | 0.0 |
| Selenocompound metabolism | 0.1 | 1.1 | 0.6 | 0.0 |

revealed considerable overlap between those two groups with low accuracy (accuracy = 0.63, R2 = 0.47, Q2 = 0.10 at the 1$^{st}$ component). While oPLS-DA over performed PLS-DA model, yet it produced low classification and prediction power (R2Y = 0.47, R2Y = 0.15). Applying RF on the same data obtained better and significant performance (P-value = 7.95e-06), yet with relatively low accuracy (0.92), sensitivity (0.93) and specificity (0.92) (**Fig 4A**). To test if this was an overfitting, we rerun RF model on different proportion of training and validation data

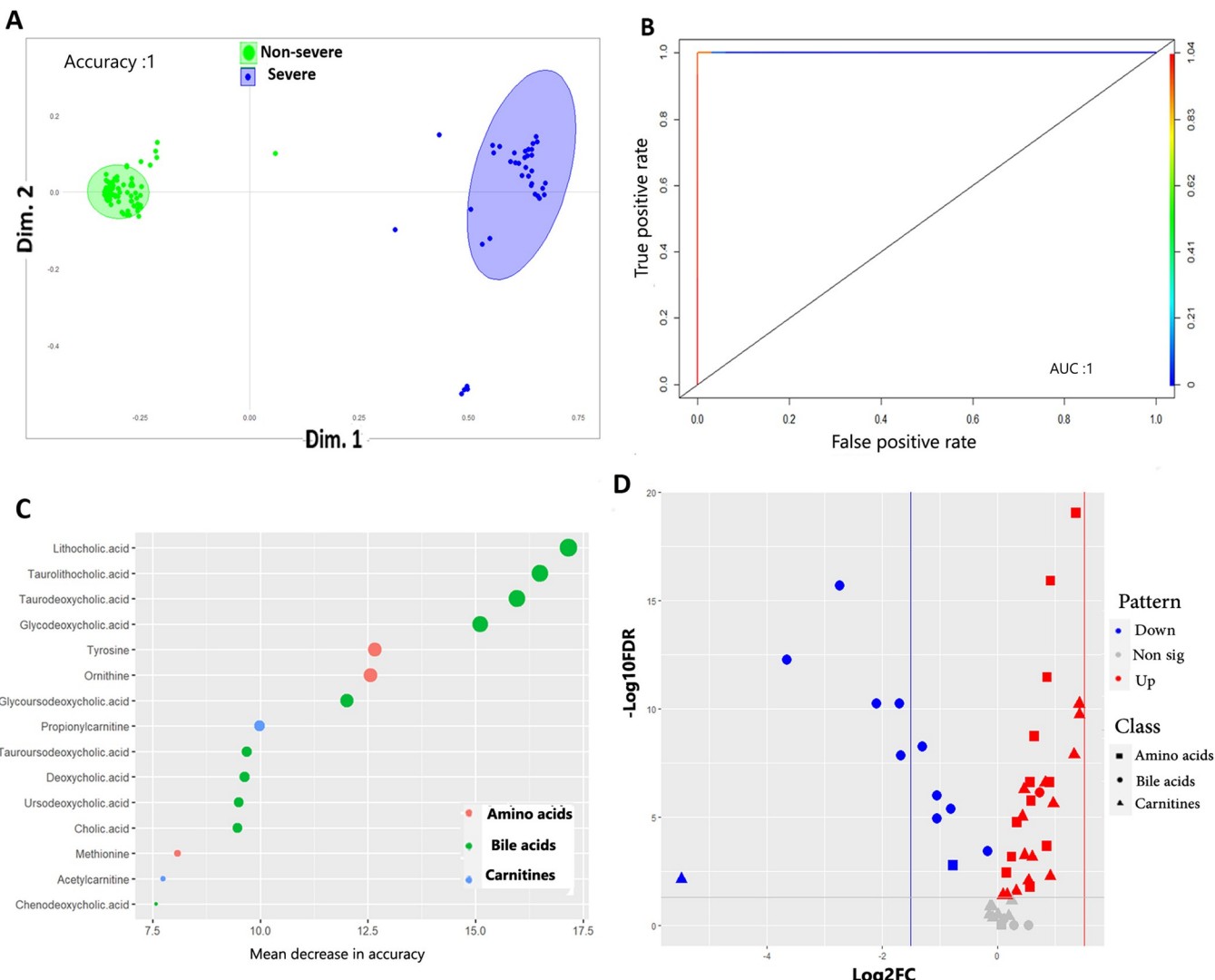

**Fig 3. Machine learning models and univariate analyses discriminating non-severe and severe COVID-19 patients. A.** Proximity plot of the RF model discriminating non-sever from severe COVID-19 patients. The ellipse shows confidence intervals **B.** ROC analyses showing the prediction ability of the model. AUC: Area under the curve. **C.** Top 15 metabolites that are important predictors for severe COVID-19 patients as revealed by RF model. The metabolites are color-grouped by their class and are ranked descending by their mean decrease in accuracy (the higher the mean decrease in accuracy the more important the metabolite). **D.** Volcano plot showing the results of the univariate analyses. The figure depicts the relationship between log2FC value of each metabolite (x-axis) against its -log10FDR (y-axis). Red and blue dots refer to metabolites that are significantly up and down regulated, respectively. Non-significant DE metabolites are shown as grey dots (-log10 adj. p value <1.3). The metabolite class is shape coded. The dashed horizontal line refers to the value of 1.3, the -log10 for a 0.05 FDR. The vertical dashed lines refer to the cutoff that equates a log2fold change value of |1.5|.

sets, which revealed similar results. ROC analyses on this model indicate a good classifier with an AUC of 0.88 (**Fig 4B**). The RF model were able to correctly predict 95.5 and 92.8% of the subjects when applied on train and test data, respectively. Given this, 4- subjects that belonged to severe patients were predicted to be critical patients by the RF model and 1 subject that was critical patient was predicted by the model as severe. Using prediction from oPLS-DA, we obtained 15 significant prognostic metabolites that have VIP score > 1 (Full list are in **S3 Table**), top of which were tauroursodeoxycholic acid, malonyl methylmalonyl succinylcarni-tine and octenoylcarnitine. Furthermore, RF model predicted list of key metabolites with high discriminatory ability according to their mean decrease in accuracy and GINI index (Full list

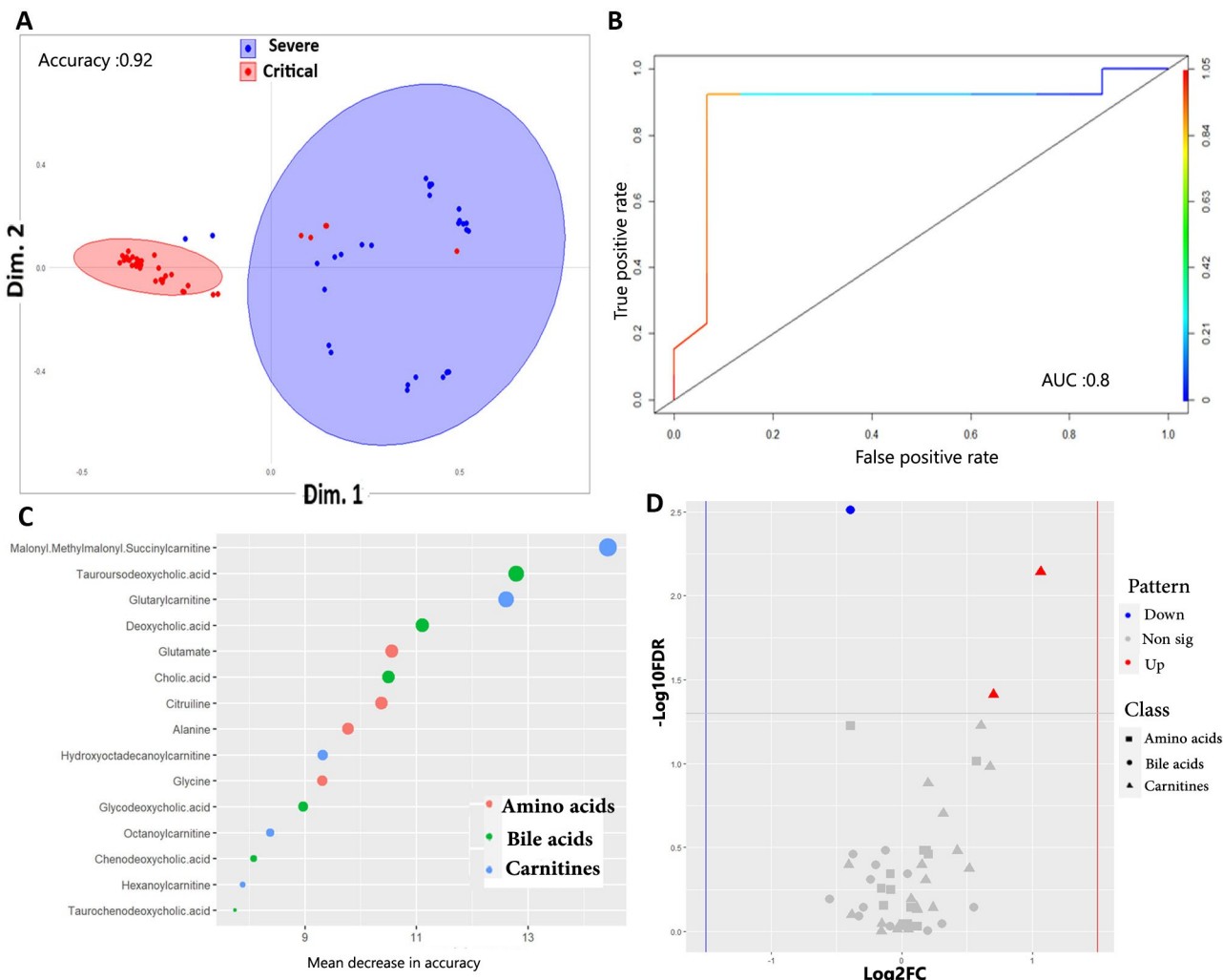

**Fig 4. Machine learning models and univariate analyses discriminating severe and critical COVID-19 patients. A.** Proximity plot of the RF model discriminating sever from critical COVID-19 patients. The ellipse shows confidence intervals **B.** ROC analyses showing the prediction ability of the model. AUC: Area under the curve. **C.** Top 15 metabolites that are important predictors for critical COVID-19 patients as revealed by RF model. The metabolites are color-grouped by their class and are ranked descending by their mean decrease in accuracy (the higher the mean decrease in accuracy the more important the metabolite). **D.** Volcano plot showing the results of the univariate analyses. The figure depicts the relationship between log2FC value of each metabolite (x-axis) against its -log10FDR (y-axis). Red and blue dots refer to metabolites that are significant up and down regulated, respectively. Non-significant DE metabolites are shown as grey dots (-log10 adj. p value <1.3). The metabolite class are shape-coded. The dashed horizontal line refers to the value of 1.3, the -log10 for a 0.05 FDR. The vertical dashed lines refer to the cutoff that equates a log2fold change value of | 1.5|.

are shown in **S4 Table**). Top 15 metabolites detected by RF and ranked by mean decrease in accuracy are shown in **Fig 4C**. The highest 3 metabolites by both mean decrease in accuracy and GINI index were malonyl methylmalonyl succinylcarnitine (carnitines), tauroursodeoxy-cholic acid (bile acid) and glutarylcarnitine (carnitine) (**S4 Table**). For further exploration, the univariate analyses performed on the same pairwise comparison identified only 3 metabolites (6%) as significantly DE between severe and critical subjects, 2 of which are up-regulated (bile acids) and 1 carnitine is down regulated (Full list are in **S5 Table** and **Fig 4D**). Combining the lists of significant metabolites that are predicted by oPLS-DA, RF models and univariate analyses obtained a panel of 2 metabolites that are important predictors for critical disease (Panel 3) (**Table 3** and **S9 Fig**).

With the reduced classification ability and accuracy of the best-applied model (i.e. RF model) in discriminating severe and critical cases, we thought to investigate if modeling only the top 15 metabolites revealed by the RF model (Those that are shown in **Fig 4C**) would enhance the model classification and accuracy. Training and validating the RF on this subset showed no additional improvement as evidenced by model accuracy (0.92) and ROC-based predictability (AUC = 0.88) (**S11 Fig**). The pathway enrichment analyses done on metabolites of panel 3 as well as that done on the top 15 metabolites identified by RF did not reveal any significant pathway, but in the latter case, aminoacyl-tRNA biosynthesis was the top enriched pathway.

**3.3.3 Models to predict COVID-19 infection outcomes (distinguishing survivors from dead subjects).** Given the information on the disease outcome, it was possible to compare patients who survived the infection and those who dead by the end of disease course. As shown in **S12 Fig**, PLS-DA model revealed considerable overlap between survived and dead patients (accuracy: 0.74, R2: 0.39, Q2: 0.21 at $1^{st}$ component). Training the data using oPLS-DA model obtained rather better separation between the two group but its performance was low (R2Y = 0.39, Q2Y = 0.27). Running RF model on the same data performed the best with a significant accuracy of 0.96, 100% sensitivity and 0.94% specificity (**Fig 5A**). The predictive ability of this model as classifier was high as determined by ROC analyses (AUC = 0.9) (**Fig 5B**).

Applying the oPLS-DA model on survived and dead subjects uncover 20 metabolites with VIP score > 1 (full list are in **S3 Table**), top of which were deoxycholic acid, prop. /Acety. ratio and hexanoylcarnitine. In addition, RF models obtained key metabolites with high discriminatory ability as determined by their mean decrease in accuracy (Full list are shown in **S4 Table**). Top 15 metabolites detected by this model ranked by mean decrease in accuracy are visualized in **S13 Fig**. The metabolites with the highest mean decrease in accuracy were deoxycholic acid, glutarylcarnitine, and hexanoylcarnitine. The univariate analyses identified 7 metabolites (14%) as significant DE between the survivor and the dead groups (full list are in **S5 Table** and S13B Fig), of which 3 were up regulated (2 bile acids and one carnitine) and 4 carnitines were down regulated. Combining the predictions from oPLS-DA, RF models and

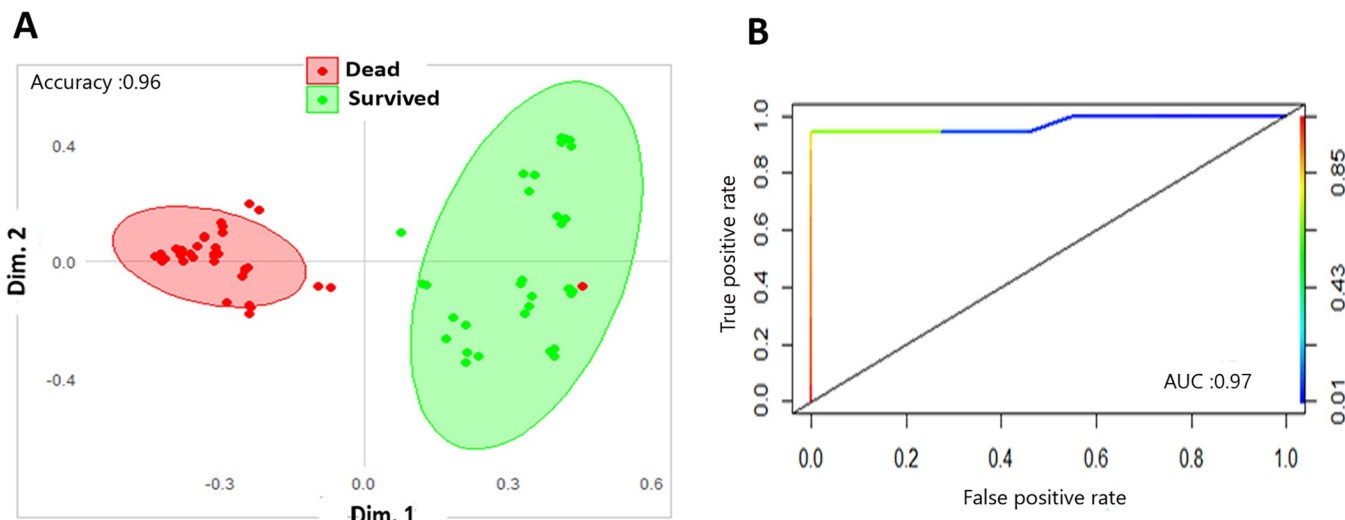

**Fig 5. Random forest classification model predicting the classification of different COVID-19 outcomes. A.** Proximity plot of the RF model discriminating survived from dead COVID-19 patients. The ellipse shows confidence intervals and each dot refers to one patient. **B.** ROC analyses showing the prediction ability of the RF model. AUC: Area under the curve.

univariate analyses produced only two metabolites that are important predictors for disease outcome (Panel 4) (Table 3).

## 3.4 Metabolic profiling in critical COVID-19 patients with and without comorbidities

Since metabolic disorders are known to be risk factors for progression of COVID-19 infection, we investigated the enrichment of diabetes and hypertension in our cohort, focusing on the critical patients group. Interestingly, PCA plot on this patient group suggests that the profile of the studied metabolites did not discriminate patients with only critical COVID-19 infection from those having critical COVID-19 infection with comorbidities. RF model run on the same data indicated a slight overlap between those having critical COVID-19 infection only from critical COVID-19 patients with comorbidities (accuracy = 0.93, P-value = 0.005). ROC analyses revealed that having hypertension alone or diabetes plus hypertension are significant predictors for critical COVID-19 infection (P-value < 0.05), with moderate AUCs of 0.7 and 0.6, respectively (S14 Fig).

Given the significant proportion of critical COVID-19 patients with comorbidities (S3A Fig), it could be that the 2 predicted significant metabolite (panel 3 in Table 3) are not prognostic markers for critical COVID-19 infection only, but are rather markers for this infection superimposed with comorbidities. To investigate this further, the levels of these 2 metabolites were compared across all categories within the critical COVID-19 patients (Fig 6). The level of taurochenodeoxycholic acid did not differ significantly across groups, whereas malonyl methylmalonyl succinylcarnitine showed significant increase in patients with critical COVID-19 plus hypertension over those with critical COVID-19 alone.

## 3.5 Diagnostic and prognostic power of combined blood indices and metabolites models

We aimed to investigate whether combining measurements of blood and metabolites and use these as inputs in one model (combined model) would enhance the classification and predictability of COVID-19 infection compared to cases when only metabolites are used (single model). The results of different comparisons are detailed in Table 2, S7 and S10 Figs. Considering all subject groups, PCA showed no enhancement in the among-groups separation in the

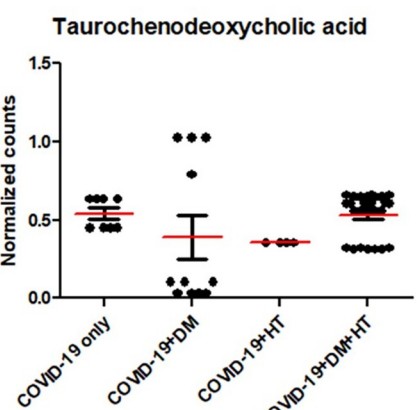
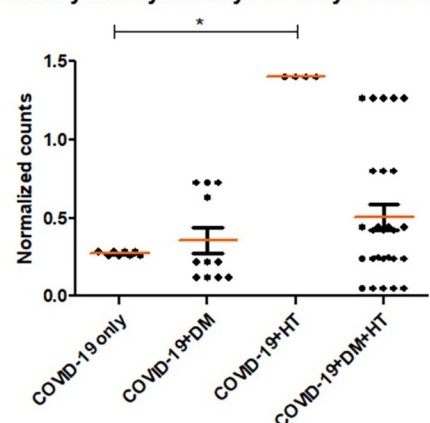

**Fig 6. Differences in levels of key metabolites in patients with only critical COVID-19 infection and those with the infection with critical COVID-19 superimposed with other comorbidities.** DM: Diabetes, HT: Hypertension.

combined model over the single one. PLS-DA as well as RF models produced similar accuracies and predictability comparing the single and combined models (**S7 Fig**). OPLS-DA model (results are denoted as 'NA' in **Table 2**) was not applicable as it is limited to pairwise comparison. Interestingly, both PLS-DA and oPLS-DA models revealed reduced accuracies, predictability and classification ability of combined model compared to single model when contrasting HC vs non-severe and non-severe vs severe groups, whereas performance of RF remained the same (**Table 2**). When comparing severe and critical COVID-19 cases, PLS-DA and oPLS-DA demonstrated an enhancement in model performance in the combined model compared to single model (**Table 2 and S10B Fig**). In particular, the accuracy, interpretability (R2) and predictability (Q2) of PLS-DA models increased by about 30.1, 14.8, 200%, respectively in the combined model. Along the same line, oPLS-DA showed an increase of 14.8 and 60% in overall variance that is explained by all features between severe and critical groups (R2Y) and the goodness of prediction (Q2Y), respectively. However, RF model showed no enhancement in the combined model over the single one with similar accuracy of 0.92, yet its predictability for combined model exhibited 10.2% increase using the AUC (AUC = 0.97) value over that of the single model (AUC = 0.88).

To more accurately identify the combination of biomarkers that would result in better classification and prediction of critical cases, we build several linear support vector machine models (each with multiple combination of these features) using a combination of the 3-important metabolites (panel 3) and the measured blood indices (n = 17) to build and compared their performance and predictive power using confusion matrices and multivariate ROC analyses. These analyses indicated that increasing number of feature combinations resulted in a gradual, yet slight enhancement in predictability and accuracies of the models. The best model was obtained when all the 20 features were compiled revealing the highest accuracy of 91% (**Fig 7A**) and the highest predictability with AUC = 0.9 (**Fig 7B**).

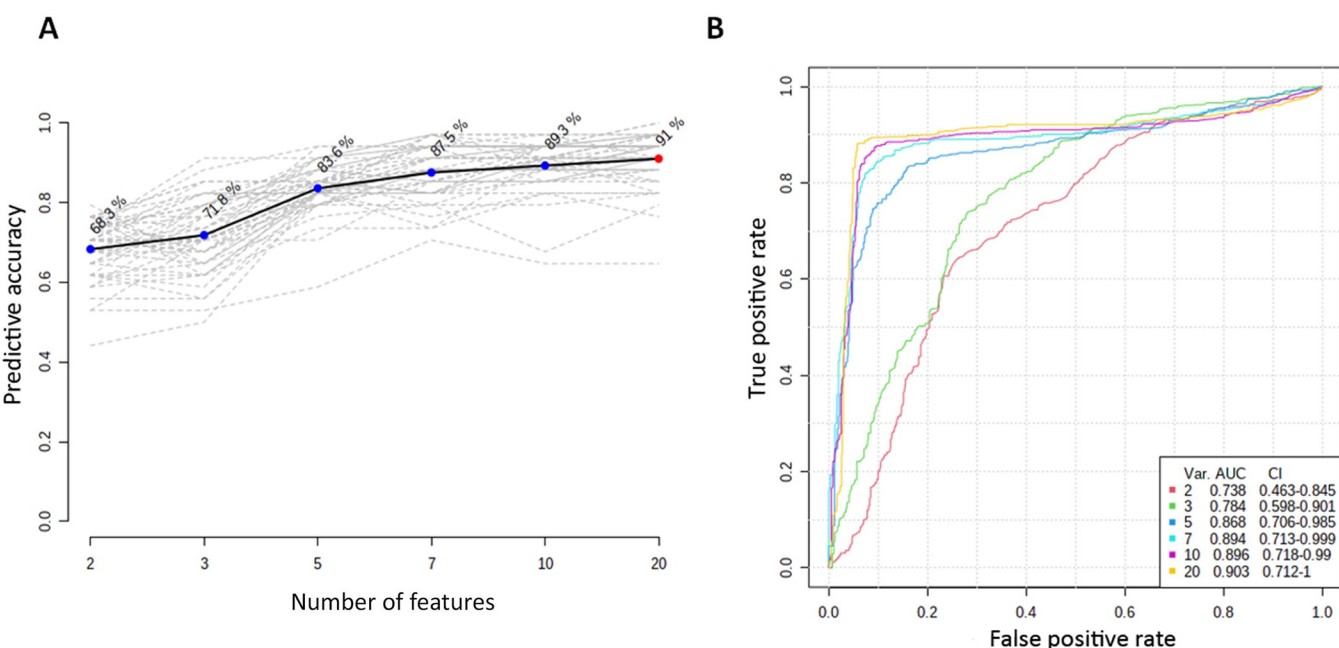

**Fig 7. Support vector machine models built using important metabolites in panel 3 (those that best discriminate severe from critical covid-19 cases) together with normalized counts of 17 blood indices. A.** Various predictive accuracies as determined by support vector machine models using feature combination (from 2–20). **B.** ROC curves showing the predictive ability of different support vector machine models of feature combination. Var. refers to different combination. AUC: Area under the curve. CI: Confidence intervals.

### 3.6 Correlation of blood parameters with key metabolites in different COVID-19 patients

Here we tried to determine if correlation between blood indices and key metabolites remains the same in different patients' group. The selection of significant correlation between pairs of metabolites and blood parameters was applied using stringent significance criteria (P-value < 0.001) and limiting this to the top 10 positive and 10 negative correlations. The analyses showed that none of the significant correlation partners that appeared in the HC was presented in other groups (**S6 Table**). Likewise, each of the COVID-19 stages showed unique pair of correlations. In particular, 80, 85 and 75% of correlation pairs appeared uniquely in non-severe, severe and critical groups, respectively.

## 4 Discussion

In this study, we aimed to investigate whether the reprogramming in metabolites that occur during COVID-19 infection could enable determining patients with varying degrees of disease severity. This would allow identifying key metabolites that are important diagnostic or prognostic biomarkers. Determining stage-specific changes in metabolites could enable informed decisions of hospital discharge or help modulating disease treatment if the infection progresses. In addition, understanding the alteration patterns in circulating metabolites during COVID-19 infection would possibly devise novel anti-SARS-CoV-2 therapeutics as shown previously [34,35].

### Demographic data of the study participants

As expected, older patients were more enriched in the group with more severe forms of the disease (i.e. those in the severe and critical stages), with significant differences in age between severe and critical cases. Age has been already known as a good predictor of COVID-19 severity [36,37]. Despite the reports about sex-induced changes in lipid, amino acid, and other metabolites, there have been reports about discrepancies over the importance of sex and age as determinants of disease occurrence and fatality [38] or non-significant gender-related differences [18]. This suggests the importance of seeking other predictors of COVID-19 occurrence and severity.

### How different is our design and analyses scheme from other research?

Machine learning models have been successfully applied on metabolomics data for predicting COVID-19 disease occurrence [11], severity and evolution [14,15,18]. However, some of these studies employed a single ML model possibly because of the inclusion of large number of patients [10,18]. In the current study, we combined the results from two common and robust ML models; the oPLS-DA and RF models, with results of univariate analyses to ensure constructing a more trusted and accurate multivariate prediction and classification scheme. Although the oPLS-DA model applied herein gave a good performance, our data indicate that the application of the non-linear more complex RF model over performed oPLS-DA (**Table 2**) in all comparisons. Indeed, RF model deems to be more suitable as its complex non-linear algorism fits the non-linear nature of most biological data [39] as stated previously [11]. RF model is known to be tolerant for outliers and is robust to over-fitting as shown on simulated and real data [40]. Using these models, our results suggest that HC were well discriminated from all other COVID-19 patients based on all metabolites or metabolite subclass indicating that the onset of COVID-19 infection is associated with strong metabolic footprint. With the

intension of determining the stage-specific alteration in metabolites, we opted to run pairwise comparisons between consecutive disease stages.

## Value of metabolites as biomarkers of early COVID-19 infection phase

In the current study, all applied ML models, and in particular the RF, showed clear discrimination between HC and non-severe COVID-19 patients even when the RF model was run on each metabolite subclass. This suggests that changes in metabolome could signal early phase of COVID-19 infection before sever disease develops. Previous data showed that metabolites change is able to distinguish COVID-19 patients from healthy subjects [15,41]. López-Hernández, Yamilé et al. showed that PCR+ non-hospitalized patients are well separated from matched controls with high accuracy of 0.88, R2: 0.8, Q2: 0.5 [37]. Similarly, Meoni et al used RF model to show a high discrimination between HC and COVID-19 patients using metabolites and lipoprotein parameters with high accuracy (0.87 to 0.91) [11]. Metabolites have shown dramatic changes at early COVID-19 infection even in the absence of clinical signs or changes in blood indices [42]. We observed that 74% of the metabolites were significant DE in the univariate analyses, with high fold change, especially malonyl methylmalonyl succinylcarnitine (4.9-fold change over HC) suggesting that the underlying reprogramming in metabolites was intense in the non-severe patients relative to HC. Our analyses showed that the differences between HC and non-severe patients originate from changes in a pool of 15 metabolites (panel 1) that are potential markers for disease initiation. Some of the metabolites from this panel were reported previously by others. For instance, methionine has been identified as a robust marker for COVID-19 occurrence in two subsequent COVID-19 waves [43]. Leucine and isoleucine were among the identified metabolites in positive COVID-19 patients, yet the difference is that leucine was down regulated in our study [44]. Our data highlights the differential regulation (mainly down-regulation) of bile acids upon COVID-19 infection in non-sever patients (except for chenodeoxycholic acid). In alignment, bile acids have been shown previously to be perturbed in COVID-19 infection and they were down regulated [45]. However, up regulation in bile acids was also observed in other studies [46,47]. The role of bile acids in COVID-19 pathogenesis has been puzzling. Generally, bile acids can limit in-vitro replication of some viruses (e.g. herpes simplex [48] and influenza A virus [49]) and promote in-vitro replication of other viruses such as hepatitis B and C [50]. Administration of antibiotics to SARS-CoV-2- infected mice resulted in reduction in certain microbiome that metabolizes primary bile acid to secondary bile acids. This leads to accumulation of primary bile acids, which subsequently were found to inhibit nsp15 endoribonuclease of the virus [51]. Furthermore, infection with SARS-CoV-2 itself causes reduction in the diversity of gut microbiome as shown in human patients [52] and primates [53] and microbiota in human gut are known to process primary bile acids into secondary bile acids such as deoxycholic acid and ursodeoxycholic/ chenodeoxycholic acid [54]. Bile acids are also biologically active molecules that organize a variety of immune functions, including inflammatory responses. Ursodeoxycholic acid does, in fact, have anti-inflammatory, antioxidant, anti-apoptotic as well as immunomodulatory properties [55]. Taken together, it is plausible to assume that the down regulation in bile acid in our study could be virus-induced to facilitate infection.

Except for malonylcarnitine (C3-DC) which showed strong up regulation, carnitines in non-severe group were down regulated. They were slightly up regulated in subsequent stages. Our data remained speculative and did not allow explaining why we observe such a variable DE pattern. Generally, L-carnitines tend to reduce inflammation and oxidative stress [56] and stimulate immunity by improving neutrophil and macrophage function [57]. Recently, it has been reported that the increase in the carnitine amount is associated with lower vulnerability

to COVID-19 [58]. Therefore, we could assume that the host triggers expression of some carnitines (e.g. malonylcarnitine) at early disease phase as a defensive mechanism, while virus tries to down-regulate other carnitine species as the disease progresses. The down regulation of arginine in the non-severe patients in our study complements previous observation in both adults and children infected with COVID-19, who had substantially decreased levels of plasma l-Arginine than controls associated with low l-Arginine-to-ornithine ratio [59].

### Value of metabolites in predicting COVID-19 severity

One core intension behind the current work was to determine how changes in metabolite could inform or explain the array of severity degrees seen in patients, and subsequently reveal prognostic biomarkers [16,41]. An obvious observation was a gradual decrease in the ability of ML models to classify or predict patients as the severity increases (**Table 2**). In parallel, the number of significant DE metabolites declined from 39 metabolites (78%), when contrasting non-severe and severe patients, to 3 metabolites (8%) when comparing severe vs critical patients. The magnitude of fold changes of these metabolites was also decreasing for most metabolites. This was also observable when analyzing metabolite subclasses. Regarding the comparison between non-severe and severe groups, our results partially agree with that reported in another study [15], where partial separation existed between symptomatic mild and more severe form of COVID-19. Similarly, PLS-DA model showed certain degree of overlap between mild (non-hospitalized) and both severe (hospitalized) plus critical (intubated) patients [37]. Of note, direct comparison among studies could be biased, especially if done on different populations because of the imminent influence of individual's genetic backup on metabolome [60], in addition to other confounders (e.g. environment and life style). The studies also differ largely in the criteria of defining severity scale of patients. Interestingly, we observed considerable overlap between severe and critical patients based on metabolite changes and only small fraction of metabolites were significant DE (n = 3). Here, the predictive ability of the RF model was the lowest (AUC = 0.8) as compared to other comparisons. Although our study did not follow up the same patient, these results suggest that changes in metabolites at the peak of COVID-19 severity might be minimal and non-reflective of patient stage of severity. Similar results were obtained by Gu et al. in China, who found low separation between sever and critical groups [15]. Our blood analyses reflected the same notion. Indeed, multiple studies have reported increased levels of inflammatory or coagulation markers such as IL-6, CRP, C-reactive proteins, procalcitonin, ferritin and D-dimers in more severe forms of COVID-19 compared to less severe ones [10,61–63]. Our data however reported that these indices increased significantly between HC, non-severe and severe patients, but not when comparing severe vs critical patients (**S2 Table**). Therefore, the inflammatory markers that were reported to be indicators for COVID-19 severity, as well as the studied metabolites, exhibited minimal differences between patients in severe and critical groups. The profile of clinical symptoms was also matched between these two groups (**S4F Fig**). Taken together, this suggests that the difference in the magnitude of changes in metabolites and inflammatory blood indices between severe and critical cases is minimal, pointing out that other biomarkers might be worth studying at that peak of disease severity, where additional hospitalization care (e.g. intubation) might be needed [37].

The analyses of the pathways suggest unique operating mechanisms in critical phase. For instance, aminoacyl-tRNA biosynthesis was significantly enriched in critical patients. Mining of transcriptomic and proteomic database of aminoacyl tRNA synthetases (aaRSa), essential enzymes in protein translation, revealed an overexpression of many aaRSa in response to infection with three SARS-CoV-2 viruses and that there is a physical interaction between virus

M protein and members of these enzymes [64]. In addition, arginine biosynthesis pathway was highly enriched in this group. Generally, l-Arginine levels have been shown to affect T cell function [65,66]. These results are supported by previous studies, which found a reduced proliferation of lymphocytes in critically ill septic patients, which has been linked to a decrease in l-Arginine availability [65].

## Combination of metabolites and blood parameters may enhance the stratification of critical COVID-19 patients

Given the low performance of metabolite model in predicting critical patients and the reported correlation between changes in metabolites and blood parameters [10,11], we sought to test whether addition of blood measurements to the metabolites (combined model) would by any means enhance the identification of critical COVID-19 patients. Similar ideas have been done by our group in the context of COVID-19 diagnosis [67]. Indeed, addition of blood parameters to metabolites (combined model) enhanced the predictive and classification power of the metabolites model (single model) for stratifying critical patients. It was also found that the addition of more blood parameters could enhance the performance of model and by extension provide better predictor panel. Similar results were obtained previously by Sindelar et al. [10], who found increased prediction ability of metabolite plus blood model (AUC = 0.7) compared to metabolite model (AUC = 0.6). This suggests the additional clinical value of measuring blood parameters during progression of COVID-19 infection.

We acknowledge that our study has some limitations. The cross-section nature of this study does not allow following up the same patients as they progress through different disease stages, which would have rather given a more precise snapshot of metabolic changes over the disease course. Indeed, doing so is challenging during the time of pandemic. Due to financial limitations, we were not able to study other blood and urine metabolites such as sphingolipids and organic acids. It is worth noting that some of our critically ill patients were either treated at home or were admitted from other clinics making it possible that prior medications given to those patients could have introduced some bias in the measured metabolites. While our study exemplifies how large sample size would allow convenient ML model construction in targeted metabolites-based investigations, a non-targeted screening of these molecules is highly warranted since it offers a complete picture of metabolite reprogramming and enable discovering novel molecules. It is also recommended to include additional patient metadata, especially the underlying metabolic disorders, when it comes to studying metabolic alterations associated with COVID-19 infection.

## 5 Conclusion

In conclusion, the underlying changes in metabolites were more characteristic, and thus could be important predictors for patients in non-severe and severe stages, but not for those suffering critical disease. Concurrent measurements of blood parameters and key metabolites could enhance the prediction ability of metabolites in those critically ill patients. Our analyses scheme suggests panels of key metabolites that could be used as diagnostic and prognostic markers for COVID-19 infection. Subsequent wide scale validation studies could further consolidate these results and open the door for using them in clinical settings.

## Supporting information

**S1 Fig. Box plot with whisker showing the mean of normalized counts of the analyzed features in each subjects in various group.** Each blue dot refers to mean normalized count of the respective metabolite subclass in one subject. The horizontal red line refers to the link between

the median of the mean normalized counts across all groups and indicates the trends of change across multiple subject's groups. Y. axes show the mean of normalized counts of metabolites in each subject.
(TIF)

**S2 Fig. Demographic, comorbidities and clinical symptoms of healthy controls and COVID-19 patients grouped by their infection severity. A.** Age (years) of the study participants. Each dot refer to one patient. The significant differences among groups were calculated using one-way ANOVA with post-hock test at a cutoff P-value of 0.05. **B.** Distribution of sex in all participants. M: Male, F: Female. **C-E.** Proportions of COVID-19 patients that show respective comorbidity (diabetes, hypertension or both). HTN: Hypertension, DM: Diabetes. **F.** Proportion of COVID-19 patients showing different symptoms.
(TIF)

**S3 Fig.** Frequency of occurrence of participants with and without comorbidities in patients group **(A)** and those with certain outcome **(B)**. The figure shows proportions of respective class as a part of the total number of the patients within respective COVID-19 severity group.
(TIF)

**S4 Fig. A & B.** Values of blood parameters in controls and COVID-19 patients grouped by their disease severity. Each dot refers to one participant. Details of numerical values and statistical differences among groups for the laboratory parameters are shown in **S2 Table.**
(TIF)

**S5 Fig.** PCA plot of the study groups (shown as color-coded circles) based on the normalized counts of metabolites in all subjects (n = 295) **(A)** and metabolites + blood parameters concentration **(B).** The distance between points are the Euclidean distance.
(TIF)

**S6 Fig. PCA plot of the study groups (shown as color-coded circles) based on the concentration of each metabolite subclass (i.e. amino acid, bile acids, carnitines) and blood indices (n = 17).** The distance between points are the Euclidean distance.
(TIF)

**S7 Fig.** Score scatter plot of PLS-DA model showing the classification of patients in all groups based on the normalized concentration of all metabolites only (n = 50) **(A)** and based on normalized concentration of both metabolites and blood **(B).** Parameters for model evaluation are shown as accuracy, variation between classes (R2Y) and predictive ability (Q2Y).
(TIF)

**S8 Fig.** Score scatter plot of PLS-DA and oPLS-DA models comparing HC vs non-severe subjects **(A)** and non-severe vs severe patients **(B)**. Each dot refers to one patient.
(TIF)

**S9 Fig. Venn diagrams showing the most important metabolites (features) as revealed by the overlap of two machine-learning models (oPLS-DA and random forest) and the univariate analyses.** The middle intersection among the 3-approaches refers to panel 1, panel 2 and panel 3, details of which are shown in **Table 3**.
(TIF)

**S10 Fig.** Score scatter plot of PLS-DA and oPLS-DA models discriminating severe and critical COVID-19 cases based on normalized concentrations of metabolites **(A)** and both metabolites

and blood indices **(B)**. Each dot refers to one patient.
(TIF)

**S11 Fig. Performance of random forest model in classifying severe and critical COVID-19 patients using the top 15 metabolites that are previously revealed by the model. A.** Score scatter plot showing the classification of both severe and critical. **B.** ROC analyses showing the predictability of the model as a classifier. AUC: Area under the curve.
(TIF)

**S12 Fig.** Score scatter plots showing the PLS-DA **(A)** and oPLS-DA models **(B)** using the normalized concentration of metabolites to classify patient's outcomes (survived and dead). Each dot refers to one patient.
(TIF)

**S13 Fig. A.** Top 15 metabolites that are important predictors for patient outcome as revealed by RF model. The metabolites are color-grouped by their class and are ranked descending by their mean decrease in accuracy (the higher the mean decrease in accuracy the more important the metabolite). **B.** Volcano plot showing the results of the univariate analyses. The figure depicts the relationship between log2FC value of each metabolite (x-axis) against its -log10FDR (y-axis). The pattern of differential expression of each metabolite are color-coded and their class are shape coded. The dashed horizontal line refers to 1.3, the–log10 for a 0.05 FDR. The vertical dashed lines refer to the cutoff that equates to a fold change value of |1.5|.
(TIF)

**S14 Fig. ROC analyses showing the classification ability of comorbidities in discriminating different severity groups of COVID-19 patients.** AUC: Area under the curve.
(TIF)

**S1 Table. Normalized concentration of metabolites analyzed in the study measured in different patients' group.**
(XLSX)

**S2 Table. A. Laboratory parameters of HC and COVID-19 patients of various severity at admission. P-value was calculated using non-parametric Kruskal Wallis test. B.** Significance levels of Pairwise comparison in laboratory parameters between various groups. Red text indicates significant comparisons.
(XLSX)

**S3 Table. Top predictor metabolites discriminating various stages of COVID-19 infection as revealed by oPLS-DA model.** VIP: Variable importance in projection.
(XLSX)

**S4 Table. Ranks of different metabolites in pairwise comparisons as revealed by random forest model.**
(XLSX)

**S5 Table. Univariate analyses of all metabolites comparing different groups of participants.**
(XLSX)

**S6 Table. Top 20 Significant correlation between pairs of metabolites and blood parameters (Significance was determined by P-value < 0.001) in different patient groups.** R is the correlation coefficient.
(XLSX)

## Acknowledgments

The authors would like to thank members of staff at the Clinical Biochemistry and Molecular Diagnostics Department, National Liver Institute, for their support.

## Author Contributions

**Conceptualization:** Gamalat A. Elgedawy, Naglaa S. Elabd, Hala H. Elsaid, Marwa L. Helal.

**Data curation:** Gamalat A. Elgedawy, Mohamed Samir, Naglaa S. Elabd, Hala H. Elsaid, Mohamed Enar, Radwa H. Salem, Randa M. Seddik, Marwa M. Omar, Marwa L. Helal.

**Formal analysis:** Mohamed Samir, Randa M. Seddik, Marwa L. Helal.

**Funding acquisition:** Mohamed Samir.

**Investigation:** Gamalat A. Elgedawy, Radwa H. Salem, Belal A. Montaser, Hind S. AboShabaan, Marwa L. Helal.

**Methodology:** Gamalat A. Elgedawy, Hala H. Elsaid, Belal A. Montaser, Hind S. AboShabaan, Shimaa M. El-Askaeri, Marwa L. Helal.

**Resources:** Naglaa S. Elabd, Radwa H. Salem, Randa M. Seddik, Marwa M. Omar, Marwa L. Helal.

**Software:** Mohamed Samir.

**Supervision:** Mohamed Samir, Shimaa M. El-Askaeri.

**Validation:** Gamalat A. Elgedawy, Mohamed Samir, Naglaa S. Elabd, Hala H. Elsaid, Mohamed Enar, Radwa H. Salem, Belal A. Montaser, Hind S. AboShabaan, Randa M. Seddik, Shimaa M. El-Askaeri, Marwa M. Omar.

**Visualization:** Gamalat A. Elgedawy, Naglaa S. Elabd.

**Writing – original draft:** Gamalat A. Elgedawy, Mohamed Samir, Naglaa S. Elabd, Hala H. Elsaid, Mohamed Enar, Belal A. Montaser, Hind S. AboShabaan, Shimaa M. El-Askaeri, Marwa M. Omar, Marwa L. Helal.

**Writing – review & editing:** Mohamed Samir, Naglaa S. Elabd, Randa M. Seddik.

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
