## [Decision Letter · Decision Letter 0]

13 Feb 2024

PONE-D-23-40352Metabolic profiling during COVID-19 infection in humans: Identification of potential biomarkers for occurrence, severity and outcomesPLOS ONE

Dear Dr. Elabd,

Thank you for submitting your manuscript to PLOS ONE. After careful consideration, we feel that it has merit but does not fully meet PLOS ONE’s publication criteria as it currently stands. Therefore, we invite you to submit a revised version of the manuscript that addresses the points raised during the review process.

We look forward to receiving your revised manuscript.

Kind regards,

Anil Bhatia, Ph.D

Academic Editor

PLOS ONE

Journal Requirements:

2. Please ensure that you have specified a) Did participants provide their written or verbal informed consent to participate in this study?

"The UPLC MS/MS instrument was a grant from Science and Technology Development Fund (STDF), Egypt. Grant number: N2338."

Reviewers' comments:

Reviewer's Responses to Questions

**Comments to the Author**

1. Is the manuscript technically sound, and do the data support the conclusions?

Reviewer #1: Partly

Reviewer #2: Yes

Reviewer #3: Yes

Reviewer #4: Partly

2. Has the statistical analysis been performed appropriately and rigorously? 

Reviewer #1: Yes

Reviewer #2: Yes

Reviewer #3: Yes

Reviewer #4: Yes

3. Have the authors made all data underlying the findings in their manuscript fully available?

Reviewer #1: Yes

Reviewer #2: Yes

Reviewer #3: Yes

Reviewer #4: Yes

4. Is the manuscript presented in an intelligible fashion and written in standard English?

Reviewer #1: Yes

Reviewer #2: Yes

Reviewer #3: Yes

Reviewer #4: Yes

5. Review Comments to the Author

Reviewer #1: “Metabolic profiling during COVID-19 infection in humans: Identification of potential biomarkers for occurrence, severity and outcomes” by Elabd and co-workers studied the metabolic profiles of different groups of patients having different severity of covid 19. I recommend the editor to accept this publication after major revision on addressing the following queries.

1. As per Methods details blood sample are collected from covid patient between the year 2021 to 2022. When they analyze these samples using LC-MS in 2023 or between 2021 to 2022 immediately collecting after the blood sample collection?

If data is not collected immediately or after 1-2 month after the sample collection. How authors will justify the change in metabolites is due to severity of the covid 19 patient not due to ageing of the sample stored in refrigeration at -80°C?? Did the authors perform any control experiment to confirm any of the metabolites they found is due to sample aging or not??

2. In Table 1, authors mention % in front of symptom (like cough/fever etc.) each group. Does this % means, among the total number of patients in each group that percent of patients are with symptoms of cough, fever etc???

Reviewer #2: The review explains all the possible metabolites from blood samples which might be helpful in prediction of the stage of COVID -19 patients and severity of the disease. I have some concerns listed below:

1. When the authors categories the various stages of COVID-19 patients (section 2.1), They wrote that severity is considered with any of the following include low oxygen saturation level which they defined below 90 and respiratory distress whereas Critical Patients who need the mechanical ventilation.

My concern is that some patients might have low oxygen saturation and does use all CPAP and other oxygen flow ways. During that stage, some patients need more oxygen flow from outside and some might need less depending upon the lungs functioning. How can we compare when the patients are in early stages of virus and need less outside oxygen flow (may be 4 or 6) versus patients in late stages of disease and need more oxygen flow (may be 14). The latter patients might also be similar to critical ones.

Secondly, sometime during the ventilation, there is ventilated pneumonia associated with disease, how that can be ruled out.

2. What age group is used during data collection? As per I understand, Is the age graph is one which represented in supporting figure S2, if that is true, how can we compare metabolites samples from age group 20-30 years versus age group 70-80 years? Please comment on that.

3. During patient treatment on severity and critical stages of disease, there are many medications given to patients. How can we rule out some metabolites might alter in the blood because of that medication. Please comment.

Reviewer #3: Manuscript titled as "Metabolic profiling during COVID-19 infection in humans: Identification of potential

biomarkers for occurrence, severity and outcomes"

I would like to congratulate the authors for such a nice work.

Is written very well with consideration of all the possible analysis.

This leads to a great addition of data set in the field of this study.

As the results suggest a greater possibility for future analysis through experimentalist using the results suggested in this Manuscript.

I found all the analysis relevant and results were interpreted very nicely.

So I found the manuscript in acceptable state.

Reviewer #4: 1. Section 2.2-Sample Collection:

Please specify the type/kind of vacutainer used, in text.

2. Section 4- Discussion:

Page 2, Paragraph 1.. "Antibiotic administered..... acid [54]."

Please review the statement for accuracy and rephrase

if required. Does the infection cause the reduction

in gut microbiome diversity or the antibiotics?

Further, please change nsp15 endonuclease to nsp15 endoribonuclease.

3. The manuscript requires an overall editing in

terms of consistency and grammar for tenses. Also for

sub/super-scripts.

6. PLOS authors have the option to publish the peer review history of their article (what does this mean?). If published, this will include your full peer review and any attached files.

Reviewer #1: No

Reviewer #2: **Yes: **Anupreet Kaur

Reviewer #3: **Yes: **Deepika Rai

Reviewer #4: No

---

## [Author Response · Author response to Decision Letter 0]

8 Mar 2024

All required revisions are uploaded in response to reviewer file.

---

## [Decision Letter · Decision Letter 1]

16 Apr 2024

Metabolic profiling during COVID-19 infection in humans: Identification of potential biomarkers for occurrence, severity and outcomes using machine learning.

PONE-D-23-40352R1

Dear Dr. Elabd,

We’re pleased to inform you that your manuscript has been judged scientifically suitable for publication and will be formally accepted for publication once it meets all outstanding technical requirements.

Kind regards,

Anil Bhatia, Ph.D

Academic Editor

PLOS ONE

Additional Editor Comments (optional):

Reviewers' comments:

Reviewer's Responses to Questions

**Comments to the Author**

1. If the authors have adequately addressed your comments raised in a previous round of review and you feel that this manuscript is now acceptable for publication, you may indicate that here to bypass the “Comments to the Author” section, enter your conflict of interest statement in the “Confidential to Editor” section, and submit your "Accept" recommendation.

Reviewer #1: All comments have been addressed

Reviewer #2: All comments have been addressed

2. Is the manuscript technically sound, and do the data support the conclusions?

Reviewer #1: Yes

Reviewer #2: Yes

3. Has the statistical analysis been performed appropriately and rigorously? 

Reviewer #1: Yes

Reviewer #2: Yes

4. Have the authors made all data underlying the findings in their manuscript fully available?

Reviewer #1: Yes

Reviewer #2: Yes

5. Is the manuscript presented in an intelligible fashion and written in standard English?

Reviewer #1: Yes

Reviewer #2: Yes

6. Review Comments to the Author

Reviewer #1: The authors have carefully addressed al queries. I recommend that the editor should accept the publication without any additional revision.

Reviewer #2: (No Response)

7. PLOS authors have the option to publish the peer review history of their article (what does this mean?). If published, this will include your full peer review and any attached files.

Reviewer #1: **Yes: **Sandeep Kumar

Reviewer #2: **Yes: **anupreet kaur

---

## [Editor Report · Acceptance letter]

9 May 2024

PONE-D-23-40352R1 

PLOS ONE

Dear Dr. Elabd, 

I'm pleased to inform you that your manuscript has been deemed suitable for publication in PLOS ONE. Congratulations! Your manuscript is now being handed over to our production team.

Kind regards, 

on behalf of

Dr. Anil Bhatia 

Academic Editor

PLOS ONE